# Assessment and Monitoring of Nail Psoriasis with Ultra-High Frequency Ultrasound: Preliminary Results

**DOI:** 10.3390/diagnostics13162716

**Published:** 2023-08-21

**Authors:** Alessandra Michelucci, Valentina Dini, Giorgia Salvia, Giammarco Granieri, Flavia Manzo Margiotta, Salvatore Panduri, Riccardo Morganti, Marco Romanelli

**Affiliations:** 1Department of Dermatology, University of Pisa, 56126 Pisa, Italy; alessandra.michelucci@gmail.com (A.M.); giorgia.salvia2@gmail.com (G.S.); giammarcogranieri@gmail.com (G.G.); manzomargiottaflavia@gmail.com (F.M.M.); salvatore.panduri@ao.pisa.toscana.it (S.P.); marco.romanelli@unipi.it (M.R.); 2Statistical Support to Clinical Trials Department, University of Pisa, 56126 Pisa, Italy; r.morganti@ao-pisa.toscana.it

**Keywords:** psoriasis, ultra-high-frequency ultrasound, onycopathy, monoclonal antibodies

## Abstract

Psoriatic onychopathy is one of the clinical presentations of psoriasis and a well-known risk factor for the development of psoriatic arthritis. High-frequency ultrasounds (HFUS > 20 MHz) have recently been used to evaluate the nail apparatus of healthy and psoriatic subjects. The aim of our study was to detect by means of ultra-high-frequency ultrasound (UHFUS 70–100 MHz) alterations of the nail bed and matrix in patients with psoriatic onychopathy and to monitor these parameters during the treatment with monoclonal antibody (mAb). We enrolled 10 patients with psoriatic onychopathy and naive to previous biologic therapies. Patients were evaluated at baseline, after 1 month and after 3 months from the beginning of mAb therapy by a complete clinical assessment and US evaluation. A UHFUS examination with a 70 MHz probe was performed on the thumbnail (I), the index fingernail (II) and the nail with greater clinical impairment (W). The following measurements were analyzed: nail plate thickness (A), nail bed thickness (B), nail insertion length (C), nail matrix length (D) and nail matrix thickness (E). Among the various parameters analyzed, some measures showed a statistically significant decrease with *p*-value < 0.05 (t0 WA = 0.52 mm vs. t2 WA = 0.42 mm; t0 WB = 2.8 mm vs. t2 WB = 2.4 mm; t0 WE = 0.76 mm vs. t2 WE = 0.64 mm; t0 IIA = 0.49 mm vs. t2 IIA = 0.39 mm). In conclusion, UHFUS could represent a viable imaging technique for the real-time evaluation and monitoring of psoriatic onychopathy, thus supporting the clinical parameters and revealing any subclinical signs of early drug response.

## 1. Introduction

Psoriasis is a chronic multisystemic and polymorphous inflammatory disease. In addition to the classic cutaneous presentation, represented by erythematous–desquamative lesions, the nail and musculoskeletal system could also be involved. The nail impairment can represent the exclusive clinical manifestation of the disease or, more frequently, it can be associated with cutaneous involvement [1,2]. Moreover, psoriatic onycopathy is a recognized risk factor for the development of psoriatic arthritis in patients with psoriasis. The nail apparatus is considered a link between the joint and the skin [3]. Since psoriasis often precedes psoriatic arthritis symptoms, dermatologists play a unique position in identifying psoriatic arthritis before the development of permanent joint damage [4]. Nail psoriasis severity index (NAPSI) is a clinical, standardized index used for quantifying the severity of nail psoriasis, analyzing nail matrix and bed alterations. Lesions involving the matrix are pitting, leuconichia, friability of the nail plate and red spots on the nail lunula. Lesions of the nail bed, on the other hand, are the oil-drop stain, onycholysis, subungual hyperkeratosis, splinter hemorrhages, Beau’s lines and trachyonychia [5]. Although it is not a validated index, it remains the reference system in clinical trials regarding psoriatic onychopathy [6].

To date, the use of ultrasound (US) in the research field of psoriasis has focused on the role of US in the study of joints, tendons and entheses of patients with psoriatic arthritis. The development of devices provided with high-resolution probes and highly sensitive power Doppler (PD) allow detailed study of tissue morphostructural features and accurate assessment of minute changes in blood flow [7,8,9].

However, more recently, US imaging has been widely used for the evaluation of nail unit features [10]. Therefore, the use of a high-frequency ultrasound (HFUS), with >20 MHz probe, has also been employed in the descriptive analysis of various nail diseases, including psoriatic onychopathy [11,12]. In the early stages of psoriatic onychopathy, a loss of the typical hyperechogenicity of the ventral portion of the nail plate can be observed. As the disease progresses, more pronounced and distinctive changes become evident in the US images. In the more advanced stages of psoriatic onychopathy, the trilaminar aspect of the nail plate, which is characteristic of healthy nails, becomes completely lost. This profound transformation is a clear indicator of the severity of the disease and reflects the extent of nail involvement: in more advanced stages of disease, the nail plate appears as a thickened, wavy, hyperechoic and inhomogeneous layer [13,14].

The use of a ultra-high-frequency ultrasonography (UHFUS) with a 70 MHz probe allows to examine the more superficial cutaneous and adnexal features with a spatial resolution in the order of 30 μm, thus offering new capabilities for the exploration of different cutaneous and non-cutaneous districts, including nails and oral mucosa [15,16,17,18]. There are no data in the literature regarding an evaluation of the nail system of patients with psoriatic onychopathy by UHFUS. Nail psoriasis represents a difficult-to-treat site for the clinician; the various options available in addition to topical therapy include the use of conventional systemic therapy (systemic retinoids, methotrexate and cyclosporine) or biologics and synthetic targeted disease-modifying drugs. Biologics, such as tumor necrosis factor (TNF)-alpha inhibitors, Interleukin (IL)-17 inhibitors and IL-23 inhibitors, have demonstrated remarkable efficacy in treating psoriatic nail disease by specifically targeting key inflammatory pathways involved in the condition. These targeted therapies offer a more favorable safety profile compared to conventional systemic agents, as they are designed to selectively interfere with the underlying disease process while minimizing the impact on the immune system [19,20]. To date, monitoring of treatment response is based exclusively on clinical assessment; however, US could represent a repeatable, noninvasive imaging technique that can be used for objective assessment of treatment efficacy.

The aim of the study was to evaluate the role of UHFUS in the assessment of psoriatic onychopathy and in the therapeutic response of naive patients treated with biological therapy compared to clinical score and patient’s quality of life. Finding a correlation between nail dystrophies, evaluated by NAPSI and US findings means developing a new and more objective way to identify and quantify the presence of nail involvement before its clinical appearance.

Moreover, UHFUS imaging could monitor US nail features improvement during the treatment and reveal any subclinical signs of drug effectiveness and nail psoriasis relapse.

## 2. Materials and Method

We conducted a prospective single-center study enrolling 10 patients with psoriasis and psoriatic onychopathy, in the absence of psoriatic arthritis, who started therapy with monoclonal antibodies (mAb) directed against TNF-alpha, (IL)-17 and IL-23. The patients were naive to previous conventional and biologic systemic therapies and were evaluated at baseline and after 1 month and after 3 months from the beginning of biologic therapy. At each visit, the clinical investigation was performed by a dermatologist expert in psoriasis who collected a photographic record of the patient and assessed clinical disease parameters such as the psoriasis area severity index (PASI), NAPSI, modified (m)-NAPSI calculated on the nail with major clinical alterations and dermatology life quality index (DLQI) [21,22,23]. For the US examination of the nail apparatus, a UHFUS with a 70 MHz probe was used (Vevo MD^®®^ FUJIFILM VisualSonics, Toronto, ON, Canada). UHFUS investigation was performed by a dermatologist expert in UHFUS blinded from the clinical diagnosis. The fingernail examination was performed in a seated position with hands placed on a table. The proper distance of the probe from the skin, to permit imaging of superficial structures, was maintained with an appropriate amount of gel [23]. The fingernail apparatus was assessed in B-MODE with a longitudinal section on the middle point of the lamina. For each patient, the first (I) and second fingers (II) of the right hand and the nail with the worst clinical aspect (W) were examined. The following parameters were measured three times by the same operator who performed the UHFUS examination and the average value of the three measurements was recorded (Figure 1)

Nail plate thickness: measured as the maximum distance between the dorsal and ventral hyperechoic plates of the nail (measure A).Nail bed thickness: measured as the maximum distance between the ventral plate of the nail and the edge of the phalangeal bone (measure B).Nail plate insertion: the non-visible part of the nail plate measured from its proximal point to its distal point (measure C).Nail matrix length: measured from the insertion of the nail plate to the proximal point of the matrix (measure D).Nail matrix thickness: measured at the point of maximum matrix thickness (measure E).

The target sample size was 10 patients, which provides 80% power at the 5% level of significance and an effect size equal to 0.1 between mean IIA at baseline and mean IIA at 3 months, with a standard deviation equal to 0.1.

Categorical data were described with absolute and relative (%) frequency and continuous data were summarized with mean and standard deviation. To compare repeated measures (t0, t1, t2) of the factors ANOVA for repeated measures was applied followed by multiple comparisons with the Bonferroni method. The significance was set at 0.05 and all analyses were carried out by SPSS v.28 technology.

## 3. Results

Our population consisted of 8/10 males (80%) and 2/10 females (20%), with a mean age of 51 years (39–63). In total, 7/10 patients (70%) were smokers or former smokers. The mean BMI was 26.7 (20.1–33.3). The mean time of disease was 18 years (5–31). All patients were naive to previous biologic therapies: three patients were treated with Adalimumab (anti TNF-alpha mAb); two patients started therapy with Ixekizumab (anti IL-17 mAb); four patients started therapy with Bimekizumab (anti IL-17A/F mAb); and one patient was treated with Tildrakizumab (anti IL-23 mAb). The clinical and US features evaluated at baseline (t0), after 1 month (t1) and after 3 months (t2) are shown in Table 1. At baseline, patients had a mean PASI of 17.5, a mean NAPSI of 40.6, an m-NAPSI calculated on the nail with major clinical changes of 9.6 and a DLQI of 12.6. A statistically significant (*p*-value < 0.05) improvement in the analyzed clinical parameters (mean PASI = 0.4; mean NAPSI = 22.9; mean m-NAPSI = 4.9; mean DLQI = 0) was detected after 3 months from the start of therapy. Among the various US parameters analyzed, some measures showed a statistically significant decrease with *p*-value < 0.05 (t0 WA = 0.52 mm vs. t2 WA = 0.42 mm; t0 WB = 2.8 mm vs. t2 WB = 2.4 mm; t0 WE = 0.76 mm vs. t2 WE = 0.64 mm; t0 IIA = 0.49 mm vs. t2 IIA = 0.39 mm). The other parameters showed a decreasing trend (measures IA, IB; IE, IIB, IIE) or an increasing trend (measures IC, ID, IIC, IIE, WC, WD) during the treatment. The results obtained from the comparison between repeated measures (t0, t1, t2) using multiple comparisons by the Bonferroni method are reported in Table 2.

## 4. Discussion

The diagnosis of nail psoriasis is usually clinical and the only severity index of psoriatic onychopathy is a strictly clinical and unvalidated score, called NAPSI. Our interest in researching US changes in the nail bed, lamina and matrix would allow us to identify an objective imaging score in addition to using NAPSI to assess the level of disease severity. To date, the application of US in psoriasis research has primarily concentrated on its role in investigating joints, tendons and entheses in patients with psoriatic arthritis. The advances in technology have led to the development of devices equipped with high-resolution probes and highly sensitive PD capabilities, enabling a more in-depth examination of tissue morphostructural characteristics and precise evaluation of minute changes in blood flow. Despite these significant advancements, the application of US in other aspects of psoriasis research still remains poorly investigated. With the introduction of UHFUS, attention has shifted towards exploring its usefulness in assessing nail psoriasis, with initial studies suggesting its potential to provide valuable insights into the severity and response to the treatment of psoriatic nail disease.

The healthy nail plate was described by HFUS examination (20 MHz) as two parallel hyperechogenic bands (railways sign), defined as the ventral and dorsal lamina of the nail plate. Between them, a hypoechogenic linear layer was detectable. The cuticle appeared as a proximally localized structure with echogenicity comparable to that of the ventral and dorsal lamina [24]. US changes of the nail plate in a patient with psoriatic onychopathy were detectable by a loss of echogenicity of the ventral plate in the early stage, and the involvement of the dorsal lamina with a complete loss of the trilaminar aspect in the advanced stages [25]. The qualitative severity of psoriatic nail alteration could be assessed ultrasonographically according to the classification presented by Wortsman et al.: type I was defined as focal, point-like hyperechoic involvement of the ventral plate; type II as continuous loss of the borders of the ventral plate; type III as the identification of wavy plates; and type IV as the complete loss of definition of both plates [26]. Also, in our study conducted with a UHFUS probe (70 MHz), we detected a trilaminar structure of the nail plate. However, the middle band presented a predominantly hypoechogenic and not totally anechogenic appearance, unlike studies in the literature.

The thickness of a normal plate varied between 0.3 and 0.65 mm [27]. Szymoniak-Lipska et al., in 2021, reported an average value in a population of healthy subjects of index fingernail plate thickness of 0.42 evaluated with a 20 MHz probe [24]. Gisondi et al. in 2012 identified with an 18 MHz probe that the average nail plate thickness of patients with psoriatic onychopathy and an average NAPSI of 18 was 0.9 mm and 0.82 mm for the thumbnail and the fingernail, respectively [28]. In another study, Idolazzi et al. identified, with an 18 MHz probe, an average value of 0.64 in a population of patients with psoriatic onychopathy with a mean NAPSI of 12.2 [11]. They also found a linear correlation between NAPSI and nail plate and bed thickness. Ally Essayed et al. identified the cut off for the diagnosis of nail psoriasis in a nail plate thickness above 0.63 mm and 0.61 mm for the thumb and the index finger, respectively (sensitivity 72% and 60% and specificity 70% and 88%, respectively) [29]. Our study collected data from a group of patients with a mean NAPSI of 40.6 and mNAPSI of 9.6 and reported mean values of nail plate thickness of 0.47 and 0.49 for the thumbnail and the index fingernail, respectively. Also, in our study, we found a correlation between higher mNAPSI and increased nail plate thickness: the plate thickness of the nail with major clinical changes was 0.52 mm.

The healthy nail bed appeared as a hypoechogenic structure localized between the ventral nail plate and the periosteum of the distal phalanx, with a thickness that ranged from 0.7 to 6.5 mm [27]. The study conducted by Ally Essayed et al. in 2015 identified a thickness of the nail bed above 1.85 mm and 1.89 mm, for the thumbnail bed and the index fingernail, respectively, to define nail psoriasis [29]. However, there was no agreement about the thickness of the nail bed to define a pathological condition. Another study identified nail bed thickness above 2.0 mm as a cut-off point for the diagnosis of psoriatic changes [30]. In more advanced stages, an increase in the distance between the ventral plate and the bony margin of the distal phalanx (>2.5 mm) was detectable [25]. Gisondi et al. in 2012 identified a mean thumbnail bed thickness of 2.95 mm in a population of patients with psoriatic onychopathy and a mean NAPSI of 18 [28]. Idolazzi et al. identified, with an 18 MHz probe, an average value of 2.5 in a population of patients with psoriatic onychopathy with a mean NAPSI of 12.2 [11]. Finally, it has been shown that nail bed thickness and plate thickness were higher in patients with psoriasis and psoriatic arthritis, with or without nail clinical involvement, compared to healthy nails [31,32].

Our study reported mean values of nail bed thickness of 3.1 mm and 2.8 mm for the thumbnail and the index fingernail, respectively. Compared to the previous study our results in terms of mean plate and bed thickness reported some differences that could be associated with the higher mean NAPSI and mNAPSI values of our population, as well as the use of UHFUS with greater axial resolution and the landmarks used for measurement. Establishing a correlation between nail dystrophies, as assessed by NAPSI, and US findings represent a significant advancement in the field of psoriasis research. Such a correlation could give new insight into the development of a novel and more objective scoring system for measuring disease severity related to nail involvement. Unlike traditional subjective assessments, UHFUS would provide a standardized and quantifiable measure of the extent and severity of nail psoriasis, leading to more accurate disease monitoring and treatment evaluation.

Few studies were reported in the literature regarding US monitoring of the nail system in subjects with psoriatic onychopathy undergoing systemic therapy. One study showed that 6 months of methotrexate therapy was able to reduce nail plate, bed and matrix thickness in patients with psoriatic onychopathy [33]. A second study revealed a reduction in matrix and nail bed thickness of patients with psoriatic onychopathy treated with acitretin [34]. There were no data in the literature regarding US monitoring of nail features with a 70 MHz probe in patients treated with mAb. Our study demonstrated that treatment with mAb anti-TNF-alpha and anti-IL determined a decrease in the nail plate and bed thickness. The statistically significant reduction in clinical parameters over time was associated with a statistically significant reduction in the nail plate thickness value of the index finger (t0 IIA = 0.49 mm vs. t2 IIA = 0.39 mm) and the finger with major clinical impairment (t0 WA = 0.52 mm vs. t2 WA = 0.42 mm) as well as a reduction in the nail bed measurement of the finger with major clinical alterations (t0 WB = 2.8 mm vs. t2 WB = 2.4 mm). The measures of lamina thickness and nail bed thickness of the other nail fingers also showed a tendency to decrease over time. UHFUS imaging’s role in monitoring the improvement of US nail features during systemic treatment is particularly promising. As an innovative biologic therapy, mAb anti-TNF-alpha and anti-IL have shown considerable efficacy in treating psoriasis, including nail psoriasis. UHFUS imaging could act as a valuable tool in assessing treatment response, allowing clinicians to visualize and objectively measure the changes in nail structures over time. This monitoring capability could help in personalizing therapeutic approaches to optimize patient outcomes. Additionally, the early detection of subclinical signs of drug effectiveness and potential nail psoriasis relapse with UHFUS imaging is a significant advancement. Identifying signs of treatment response or relapse earlier than traditional clinical assessments may enable prompt adjustments to therapy, ensuring patients receive the most effective and timely interventions.

The proximal nail fold was described by studies in the literature as a structure with lower echogenicity than the ventral and dorsal plates of the nail plate [24]. In contrast, examination with a 70 MHz probe allowed precise identification of the appearance of the nail plate insertion in the distal phalanx. The dorsal and ventral plates continued beyond the cuticle as two hyperechogenic bands that converged proximally with a hypoechogenic layer in the middle. The sharpness and linearity of the US bands were higher in subjects with fewer detectable clinical changes (lower NAPSI and m-NAPSI). No data regarding the measurement of nail plate insertion length in the distal phalanx were available in the literature. The data found from our study showed that the clinical improvement detectable by a reduction in NAPSI and m-NAPSI due to treatment was associated with elongation and increased linearity of the nail plate insertion (Figure 2).

It could be hypothesized that the inflammatory state of the matrix and the periungual tissues of patients with psoriatic onychopathy caused an increase in the level of mechanical compression in the nail insertion determining qualitative and quantitative US alterations. The improvement of these parameters during the treatment with anti-IL mAbs, able to reduce the inflammatory burden, would support our hypothesis, although further studies are needed to confirm these preliminary data.

The matrix was defined as a hypoechoic structure with blurred borders, near the nail fold, 1–5.3 mm long, detectable in a minority of subjects and variable in different types of fingers. The matrix was identified with more precision in the fourth and fifth fingers and more often in female patients [24]. Our study defined the nail matrix as a hypoechogenic structure that surrounded the nail insertion in the distal phalanx, whose linearity and US sharpness were higher in patients with lower nail changes at baseline and after treatment with mAb. In our study, we also measured nail matrix length reporting higher values for patients with fewer clinical alterations at baseline and after treatment with mAb. US studies in patients with nail PsO also revealed significantly increased thickness of nail matrix compared to healthy nails. In particular, Krajewska-Włodarczyk reported mean nail matrix thickness values of 1.96 mm in patients with psoriatic onychopathy obtained with a linear probe with a frequency ranging from 12 to 48 MHz. [35] The importance of an objective assessment of nail matrix impairment was related to the association between the involvement of the distal interphalangeal joint and the nail root inflammation. Psoriatic arthritis is a complex condition that involves not only the skin but also the musculoskeletal system. Recent advancements in US measurements gave new insight into the evaluation of nail alterations in subjects with psoriatic arthritis, offering valuable insights into the disease’s underlying pathophysiology. In the pathophysiology of psoriatic arthritis, the central role of TNF-α and IL-17 had been highlighted. MAbs directed against these molecules had shown potential efficacy in reducing early signs of psoriatic arthritis activity, representing a significant clinical–therapeutic challenge. Optimizing the early management of psoriasis patients through targeted biologic therapies could potentially prevent or slow down the progression of arthritis, leading to better long-term outcomes and improving the quality of life for affected individuals. Nail involvement in psoriasis patients has been recognized as a significant risk factor for developing psoriatic arthritis, emphasizing the importance of nail assessments in dermatological practice. Dermatologists play a pivotal role in the early detection of psoriatic arthritis, as psoriasis often precedes joint manifestations. Recognizing patients before permanent joint damage occurs is crucial for initiating timely and appropriate interventions to improve patient outcomes. However, the possibility that psoriatic nail disease may be related to enthesis microdamage or mechanically stressed tissues still awaits confirmation by high-resolution imaging [36]. Our study reported a mean nail matrix thickness value of 0.66 mm for the thumbnail, 0.74 mm for the index fingernail and 0.76 mm for the fingernail with major clinical impairment. A statistically significant reduction was found for matrix thickness of the nail with greater clinical changes after 3 months of therapy (t0WE 0.76 mm vs. t2WE 0.64 mm). The differences observed between our values and those presented in the literature are likely related to the differences in the population analyzed (higher NAPSI and mNAPSI), the different US frequencies used, and the landmarks chosen to perform the measurement. As in the case of nail insertion, it is possible to hypothesize that the inflammatory burden of the periungual tissues is able to cause alterations such as quantitative changes at the level of the nail matrix. The decrease in inflammatory load following treatment would reduce mechanical and inflammatory compression at the level of the matrix, leading to its elongation and decrease in thickness, as well as an improvement in its sharpness, evaluated by US examination. Finally, our study demonstrated that treatment with mAb determines an improvement in US nail features even before a reduction in the corresponding clinical index: after one month of therapy, a decreasing trend in measures A, B, E and an increasing trend in measures C and D in the absence of a change in clinical values were detected for all the fingernails analyzed.

## 5. Conclusions

Several studies in the literature reported the use of HFUS in assessing the severity of psoriatic nail disease and its correlation with clinical parameters (mNAPSI or NAPSI) following the introduction of pharmacological therapy. We presented, for the first time, the preliminary results obtained using UHFUS as an objective parameter for psoriatic onychopathy assessment and monitoring, able to support clinical evaluation in future clinical trials and studies. Moreover, UHFUS would allow early assessment of nail changes even before clinical changes are detectable. From our study, the measurements of the nail apparatus obtained through UHFUS showed a correlation with the clinical improvement of psoriatic nail disease, sometimes even anticipating it. For example, during the first month of therapy, a significant reduction in nail plate thickness was observed in the absence of an obvious clinical improvement in NAPSI and mNAPSI. Therefore, we believe that this tool can be used to identify subclinical modifications that are not directly detectable during a dermatological clinical examination. The clinical improvement becomes evident months after the start of therapy, raising the suspicion of a lack of therapeutic response. However, changes in UHFUS parameters appearing earlier could be suggestive of an initial response to pharmacological therapy and could guide the decision to maintain a particular pharmacological treatment. Furthermore, evaluating a larger number of patients on different pharmacological therapies could reveal, through intergroup analysis, which drug can provide a faster and more effective improvement in psoriatic nail disease. This would also guide the initial choice of pharmacological treatment. Finally, because of the close anatomical relationship between the enthesis and the nail root, the evidence of early US changes in patients with early psoriatic arthritis could play a role in the early management and prevention of irreversible joint damage. Nails are integral components of the musculoskeletal system, acting as a link between the joints and the integument. Therefore, understanding nail alterations could offer valuable clues about the underlying joint disease and support early diagnosis and disease management. Collaborative efforts between dermatologists, rheumatologists, and other healthcare professionals are crucial in ensuring comprehensive care for patients affected by psoriasis and psoriatic arthritis. The main limitation of this study is related to the small sample and the absence of a comparison between the different treatment outcomes. In conclusion, the incorporation of UHFUS imaging in the evaluation of nail psoriasis represents a promising step towards more precise and objective disease assessment. It has the potential to revolutionize the way we understand, diagnose, and treat nail psoriasis, ultimately leading to better patient care and improved long-term outcomes. As research in this area continues to evolve, we can expect UHFUS imaging to play a pivotal role in shaping the future of psoriasis management. Further investigations with prolonged observation time and a greater number of patients will be necessary to confirm the suggested hypotheses.

## Figures and Tables

**Figure 1 diagnostics-13-02716-f001:**
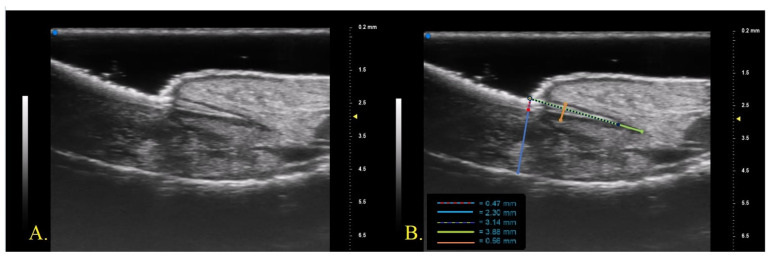
(**A**)**.** UHFUS nail morphology; (**B**). UHFUS structural features measured: nail plate thickness, red-blue dotted line (measure A); nail bed thickness, blue line (measure B); nail plate insertion, purple-green dotted line (measure C); nail matrix length, green line (measure D); nail matrix thickness, orange line (measure E).

**Figure 2 diagnostics-13-02716-f002:**
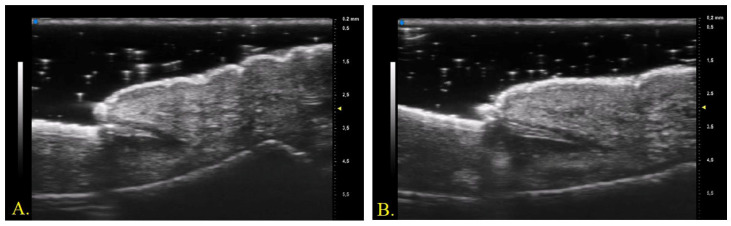
UHFUS (70 MHz) examination of nail apparatus, evaluated at baseline (**A**) and after 3 months of Ixekizumab (**B**).

**Table 1 diagnostics-13-02716-t001:** Comparison between repeated measures ANOVA. Statistics: mean (sd). Psoriasis area severity index (PASI), NAPSI, modified (m)-NAPSI, dermatology life quality index (DLQI), thumbnail (I), index fingernail (II) nail with worst clinical impairment (W), nail plate thickness (A), nail bed thickness (B), nail insertion length (C), nail matrix length (D) and nail matrix thickness (E).

Factor	t0	t1	t2	*p*-Value
PASI	17.5 (14.2)	5 (3.6)	0.4 (0.6)	0.017
NAPSI	40.6 (25.2)	40.6 (25.2)	22.9 (7.8)	0.032
mNAPSI	9.6 (4.1)	9.6 (4.1)	4.9 (1.8)	0.003
DLQI	12.6 (7.2)	3.4 (2.3)	0 (0)	<0.001
I A	0.47 (0.19)	0.46 (0.15)	0.38 (0.11)	0.109
I B	3.1 (0.7)	2.9 (1)	2.6 (0.3)	0.147
I C	3.9 (1.4)	4.3 (0.7)	4.1 (1.1)	0.506
I D	4.4 (1.3)	4.7 (0.7)	4.6 (1)	0.459
I E	0.66 (0.22)	0.61 (0.13)	0.58 (0.11)	0.236
II A	0.49 (0.14)	0.47 (0.16)	0.39 (0.07)	0.032
II B	2.8 (0.3)	2.8 (0.4)	2.6 (0.3)	0.057
II C	4.1 (1.3)	4.4 (1.1)	4.5 (1.1)	0.087
II D	4.9 (1.3)	5.3 (1.2)	5 (1.1)	0.448
II E	0.74 (0.25)	0.68 (0.19)	0.64 (0.13)	0.295
W A	0.52 (0.15)	0.48 (0.1)	0.42 (0.1)	0.039
W B	2.8 (0.3)	2.7 (0.5)	2.4 (0.3)	0.003
W C	4 (1.2)	4.2 (1.1)	4.1 (1.1)	0.275
W D	4.8 (1.1)	4.9 (1)	5.3 (1.1)	0.293
W E	0.76 (0.2)	0.71 (0.17)	0.64 (0.13)	0.025

**Table 2 diagnostics-13-02716-t002:** Multiple comparisons by Bonferroni method. Statistics: *p*-value. Psoriasis area severity index (PASI), NAPSI, modified (m)-NAPSI, dermatology life quality index (DLQI), thumbnail (I), index fingernail (II) nail with worst clinical impairment (W), nail plate thickness (A), nail bed thickness (B), nail insertion length (C), nail matrix length (D) and nail matrix thickness (E). Not evaluable (ne).

Factor	t0 vs. t1	t0 vs. t2	t1 vs. t2
PASI	0.022	0.014	0.010
NAPSI	ne	0.097	0.097
mNAPSI	ne	0.010	0.010
DLQI	0.010	0.001	0.003
II A	0.662	0.038	0.138
W A	0.478	0.054	0.061
W B	0.682	0.002	0.208
W E	0.868	0.040	0.144

## Data Availability

Data Availability Statement: The data presented in this study are available on request from the corresponding author. The data are not publicly available due to privacy reasons.

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
