# Peer review of "Assessment and Monitoring of Nail Psoriasis with Ultra-High Frequency Ultrasound: Preliminary Results"

_diagnostics, 2023, doi:10.3390/diagnostics13162716_

Round 1

Reviewer 1 Report

This is an interesting observational study and should be relevant to practicing doctors. There are some minor suggestions I would like to make:

1. Please consider revising the title by removing terms like 'new frontier' as these are strong assertion that may not be supported by the paper. I also suggest including study design into the title.

2. Please explain who performed UHFUS, were they blinded from the clinical diagnosis, did the same investigator perform all the UHFUS and whether the sonographers' performance have been validated for inter-sonographers consistency?

3. Please discuss how the paper could conclude that UHFUS is effective in assessing  nail lesions when there was no comparison with the gold standard for nail psoriasis assessment/monitoring.

Author Response

Thanks for your comments. Please see the attachment for the comments.

Reviewer 2 Report

Diagnostics

New frontiers in the assessment and monitoring of nail psoriasis: the role of Ultra-High Frequency Ultrasound

Review

·         MATERIALS AND METHODS

1.       Sample Size: With only 10 patients included in the study, the study might be underpowered to detect meaningful effects. The authors should provide some rationale for this sample size or a power analysis showing that this sample size is sufficient. While you did acknowledge the sample size as weakness, there is no rationale for it.

2.       While repeated measures ANOVA can technically be performed with a small sample size such as 10, the reliability and power of the test will be very low. A small sample size increases the risk of a Type II error and any significant results you do find with such a small sample should be interpreted with caution.

Moreover, having data drop-out at the 3rd time point further complicates the matter as it could lead to biased results if the drop-out was not completely random (i.e., if the patients who dropped out were systematically different from those who did not).

It would be highly advisable to increase the sample size to improve the statistical power and reliability of the study's results.

3.       Overall, the provided sample size in this study is quite small for this type of analysis, and results should be interpreted cautiously. The researchers should consider increasing the sample size in future studies, if possible.

4.       Measurements: It might also be beneficial to specify the reliability of these measurements where applicable.

5.       Repeated Measures ANOVA: It's not clear if they checked the assumptions of this test, such as sphericity and normality assumption of residuals. Violation of these assumptions can lead to biased results. It would be helpful if they could provide information on this.

6.       Effect Sizes: Lastly, while the authors note that significance was set at 0.05, they do not mention if they will report effect sizes. Given the small sample size, significant results might not necessarily be meaningful or practically significant. Therefore, reporting effect sizes would be helpful in interpreting the results.

·         RESULTS

1.       Participant Characteristics: There seems to be a discrepancy in the number of patients. The first line mentions that there are 10 patients, however, in the next line it is mentioned that 7 out of 13 patients are smokers. Please confirm the exact number of patients included in the study.

2.       Follow-up: It is mentioned twice that 6/10 patients reached a follow-up of 6 months after treatment initiation. This repetition seems redundant and can be avoided.

3.       Table 1: Table 1 has Table 10 caption? There is no proper heading for Table 1(0). What am I looking at? If these are results of repeated measures ANOVA you should clearly state so. You must include F-ratios, degrees of freedom and effect size measure as bare minimum.

What are the values in parentheses? SDs? If so, this should be stated.

4.       Table 2: This table is also labelled as Table 10 and there is no heading. What are the values in table? Differences? Test statistics? P-values? Effect sizes? Also, the term "ne" is not clear and may need to be defined or replaced with a more commonly accepted term.

This table should be completely overhauled with all three relevant stats, such as critical values, p-values, differences, effect sizes, sample size at current time point.

5.       Improvement in Clinical Parameters: It's mentioned that there was a significant improvement in the analysed clinical parameters after 3 months of therapy, but some of these values appear to have increased rather than decreased (e.g., NPQ10 from 27.1 at baseline to 29.5 after 3 months). Doesn’t NPQ10 range from 0 to 20 with lower scores being a better outcome? Your NPQ10 scores are higher than 20 and you seem to interpret scores as “higher is better”.
Ortonne JP, Baran R, Corvest M, Schmitt C, Voisard JJ, Taieb C. Development and validation of nail psoriasis quality of life scale (NPQ10). J Eur Acad Dermatol Venereol. 2010 Jan;24(1):22-7. doi: 10.1111/j.1468-3083.2009.03344.x. PMID: 20050290.

The study at hand fails to establish its credibility due to multiple shortcomings. First and foremost, the sample size of 10 patients is insubstantial, leaving the study severely underpowered, prone to Type II error and susceptible to biased results due to data drop-out at the third time point. The authors need to provide a solid rationale for this sample size or perform a power analysis to justify its adequacy.

Measurement reliability and adherence to the assumptions of repeated measures ANOVA, such as sphericity and normality of residuals, remain unclear, risking the introduction of bias. It is critical that the authors elucidate these points. In addition, the significance level of 0.05 is insufficient without reporting effect sizes, particularly given the small sample size.

The presentation of the results section raises serious questions about the accuracy and clarity of the data. Discrepancies in patient numbers, redundant statements about follow-up, unlabelled tables, undefined terminology, and lack of crucial statistics raise concerns about the rigor and reliability of the analysis.

Finally, the NPQ10 scale is interpreted in a manner inconsistent with existing literature, leading to questionable conclusions about improvement in clinical parameters. Until these concerns are addressed, the validity of the study remains suspect, and I would recommend rejection.

Round 2

Reviewer 1 Report

Thank you for the revision. The authors have responded to the comments and made appropriate amendments.

Reviewer 2 Report

I' m staying  at my previous review.